# Continual Learning with Dynamic Sparse Training: Exploring Algorithms for Effective Model Updates

Murat Onur Yildirim[1], Elif Ceren Gok Yildirim[1], Ghada Sokar[1], Decebal Constantin Mocanu[2], Joaquin Vanschoren[1]

[1]Eindhoven University of Technology, [2]University of Luxembourg

`m.o.yildirim@tue.nl, e.c.gok@tue.nl, g.a.z.n.sokar@tue.nl,`
`decebal.mocanu@uni.lu, j.vanschoren@tue.nl`

Continual learning (CL) refers to the ability of an intelligent system to sequentially acquire and retain knowledge from a stream of data with as little computational overhead as possible. To this end; regularization, replay, architecture, and parameter isolation approaches were introduced to the literature. Parameter isolation using a sparse network which enables to allocate distinct parts of the neural network to different tasks and also allows to share of parameters between tasks if they are similar. Dynamic Sparse Training (DST) is a prominent way to find these sparse networks and isolate them for each task. This paper is the first empirical study investigating the effect of different DST components under the CL paradigm to fill a critical research gap and shed light on the optimal configuration of DST for CL if it exists. Therefore, we perform a comprehensive study in which we investigate various DST components to find the best topology per task on well-known CIFAR100 and miniImageNet benchmarks in a task-incremental CL setup since our primary focus is to evaluate the performance of various DST criteria, rather than the process of mask selection. We found that, at a low sparsity level, Erdős-Rényi Kernel (ERK) initialization utilizes the backbone more efficiently and allows to effectively learn increments of tasks. At a high sparsity level, unless it is extreme, uniform initialization demonstrates more reliable and robust performance. In terms of growth strategy; performance is dependent on the defined initialization strategy and the extent of sparsity. Finally, adaptivity within DST components is a promising way for better continual learners.

## 1. Introduction

Continual learning (CL) is a method of developing models that can learn and adapt continuously from a stream of data, or tasks, unlike the standard batch learning that necessitates access to all categories' data simultaneously. Therefore, it allows efficient learning and avoids the storage of large amounts of data. However, this often comes at a price, known as *catastrophic forgetting*, the degradation of previously learned information as the deep learner focuses on learning new ones. Four major approaches have been explored to address this issue. Regularization [1–5] prevents abrupt changes in the neural network weights. Replay stores a few real exemplars [6–8] or generates some synthetic exemplars by learning the data distribution [9–12] per class and rehearses them. Architecture expansion [13–17] modifies the network architecture, typically by adding extra network components. Parameter isolation [18–27] partitions the neural network into subnetworks that are dedicated to different tasks and freezes the weights of subnetworks which allows to preserve previously learned knowledge. In this paper[1], we focus on the parameter isolation approach due to its fixed network capacity usage, low tendency to forget, and its memory- and computation-wise efficiency.

Parameter isolation typically freezes the connections of previous subnetworks and only updates newly added ones to be able to learn the current subnetwork. It can be performed via mask learning [18, 19], iterative pruning [20–22] or dynamic sparse training (DST) [23–27]. Mask learning utilizes a fixed, usually pre-trained, dense backbone and only intends to find the best mask for a given task. Iterative pruning trains a dense network and then iteratively removes the unpromising weights with a smallest magnitude whereas DST aims to train sparse network from scratch and find the best topology simultaneously by dropping and growing different

---

[1]code is available at: `https://github.com/muratonuryildirim/CL-with-DST`

components of the network during training to discover the best sparse topology. It reduces the number of parameters quadratically during both training and inference without compromising the accuracy [28]. DST can be performed on neuron-level or connection-level which are known as structured and unstructured pruning respectively. Although structured pruning allows computationally more efficient subnetworks, it has been shown to perform less compared to unstructured pruning on higher sparsity levels [29] which reserves more space for subsequent tasks in the network.

DST has emerged as a promising technique for enhancing the efficiency and generalization of neural networks by dynamically pruning and sparsifying network parameters during training for standard batch learning. However, when applied to CL, the standard settings of DST for batch learning may not fully exploit its potential. Investigating different components of DST under CL settings is essential for several reasons. Firstly, continual learning necessitates the retention of knowledge acquired from previous tasks, which may require different pruning and reinitialization strategies. Secondly, the optimal strategies for DST may vary in a continual learning context, where tasks arrive sequentially and more than one subnetwork should be learned incrementally, using the same backbone. Finally, continual learning scenarios often involve class imbalance, concept drift, and varying task complexities, which may require customized approaches to dynamic sparsity. Thus, by exploring and adapting the various elements of DST to continual learning, we can unlock its full potential for mitigating the challenges of CL and improving the efficiency and performance of neural networks in lifelong learning scenarios. Therefore, in this paper, we conducted large-scale experiments on CIFAR100 and miniImageNet with different backbones and different number of tasks. We analyze unstructured DST performance in CL in terms of incremental accuracy, backward, and forward transfer under various sparsity levels, initializations, and growth strategies. We focus on task-incremental CL since we specifically want to observe the performance of different DST criteria, not the process of mask selection. Our findings are:

I. At a low to moderate (80-90%) sparsity level, Erdős-Rényi Kernel (ERK) initialization utilizes the backbone more efficiently by using the connections within the narrow layers sparingly which allows to effectively learn increments of tasks.

II. At a high (95%) sparsity level, unless it is extreme (99%), uniform initialization demonstrates more robust incremental performance by maintaining stable performance over tasks.

III. From the growth strategy standpoint, performance outcomes are dependent on defined initialization strategy, and the degree of sparsity.

IV. There is no panacea yet for DST in CL and selecting the DST criteria adaptively per task holds a promise to enhance the performance, compared to the pre-defined strategies for all tasks. We show that a naive alteration between different growth strategies boosts incremental performance.

## 2. Background and Related Work

**DST for Standard Batch Learning.** Dynamic sparse training is a technique to attain a sparse network in which the model is initialized with a random sparse topology and then the connections are dynamically pruned and grown during the training process to improve computational efficiency and generalization performance [28]. This involves an update period of the network topology and a drop/growth strategy to explore the network within a pre-defined sparsity level while minimizing the loss function by preserving important weights for a given task. By reducing the number of connections, dynamic sparse training can speed up training and reduce memory requirements while maintaining or even improving model accuracy [30].

SET [28] is the pioneer study of DST and discovers an optimal sparse connectivity pattern during training by introducing *Erdős-Rényi* (ER) initialization and random growth. ER initialization assigns more connections to smaller layers while allocating fewer connections to the large ones in MultiLayer Perceptrons (MLPs) with $\frac{\epsilon(n^{l-1}+n^l)}{n^{l-1}n^l}$ where $n^l$ denotes the number of neurons at layer $l$ and $\epsilon$ indicates the sparsity level. Random growth randomly connects an equal number of weights that are dropped based on their *magnitudes* to explore a new sparse topology. Inspired by SET, RigL [31] and SNFS [32] introduced gradient growth and momentum growth: the idea of using gradient and momentum information obtained by backward pass is to grow the promising connections that accelerate the exploration of optimal sparse topology. Although gradient and momentum growth accelerate the exploration of optimal sparse topology, they require a dense backward pass

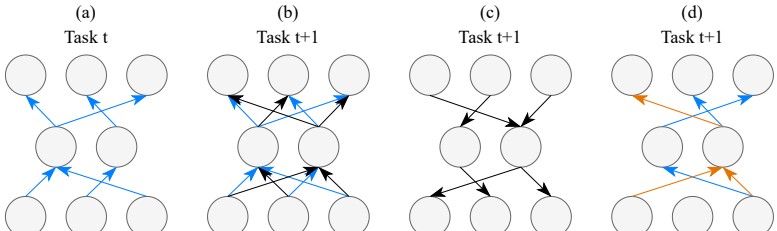

Figure 1: Illustration of Continual DST. (a) **Learned connections** for Task t, (b) in the forward pass of a new task, the model is free to use all the connections, (c) yet in the backward pass it is only allowed to update the **unassigned connections**, (d) after dynamically exploring the network with steps (b) and (c) under a sparsity level of s%, subsequent subnetworks which can include **new connections** and previously **learned connections** are obtained.

to use the information of both existing and the non-existing connections. To improve this, Top-KAST [33] opted to use the gradients corresponding to a subset of the non-existent connections. Furthermore, [34, 35] thoroughly study the dynamic sparse training and sparse topologies, and found that the performance of the sparse networks can outperform the dense counterparts.

**DST for Continual Learning.** Prior work [36] showed that dynamically shrinking and expanding the model capacity effectively avoids the excessive pruning, compared to the iterative pruning over sequential tasks. Hence, Dynamic Sparse Training is a viable Parameter Isolation approach for Continual Learning since it can assign specific components to different tasks while exploring the network with different topologies. SpaceNet [23] and AFAF [24] opt to utilize DST in a structured manner that explores the best sparse topology on a neuron-level whereas NISPA [25], WSN [26] and SparCL [27] perform it in an unstructured manner which is a connection-level search. SpaceNet allows having limited common units among subnetworks to prevent interference between tasks while NISPA, WSN, and AFAF exploit units from previous subnetworks fully to utilize the backbone efficiently and also transfer the relevant knowledge. NISPA selects the candidate connections randomly while WSN chooses them based on the gradients. To enable practical on-device CL, SparCL integrates weight sparsity with data removal and gradient sparsity.

However, in CL scenarios, none of those studies have extensively explored the effect of different initialization, drop, and growth strategies which have a high influence on the final sparse neural network topology and performance.

## 3. Method

In this section, we give an overview of DST and its application in the context of CL where the goal is to learn and retain knowledge from a sequence of tasks without catastrophic forgetting. We present the underlying formulation and optimization objectives that guide our exploration of hyperparameters and design choices within this framework.

**Overview.** As shown in Figure 1, Continual DST starts with an over-parameterized neural network in which we searched for the task-specific subnetworks and only changed the weights that are not trained on the previous tasks. In other words, we froze the parameters of the selected subnetwork after training to prevent catastrophic forgetting yet we allowed to use those frozen parameters in the subsequent tasks. This supports the transfer of previous knowledge to future tasks, also known as forward transfer. It is particularly important for large-scale continual learning problems as it decreases the time required to converge during sequential learning.

**Continual DST.** CL involves updating a neural network with a sequence of learning tasks $\mathcal{T}_{1:t} = (\mathcal{T}_1, \mathcal{T}_2, ..., \mathcal{T}_t)$, each with a corresponding dataset $\mathcal{D}_{\mathcal{T}} = (x_{i,t}, y_{i,t})^{n_t}$ consisting of $n_t$ instances per task. Upon encountering a new learning task, a deep network optimized to map the input instance to the classifier space, denoted as $f_\Theta : \mathcal{X}_t \rightarrow \mathcal{Y}_t$, where $\Theta$ stands for the parameters of the network. Note that, the learning tasks are mutually exclusive, i.e., $\mathcal{Y}_{t-1} \cap \mathcal{Y}_t = \emptyset$ and we assumed that task identity is given in both the training and the testing stage. The goal of Continual DST is to effectively learn the subnetwork by solving the optimization problem that minimizes the following function:

$$\mathcal{M}_t^* = \arg\min_{\mathcal{M}_t} \mathcal{L}(\Theta_s, n_t) = \arg\min_{\mathcal{M}_t} \sum_{i=1}^{n_t} [CE(f(x_{i,t}; \Theta_s), y_{i,t})] \tag{1}$$

Here, $\mathcal{L}(\Theta_s, n_t)$ is the loss function, which is Cross-Entropy ($CE$) in our experiments, with a pre-defined fraction of parameters $\Theta_s$, and by solving this optimization problem, mask $\mathcal{M}_t$ is determined. Note that, although sparsity per task is predefined, the overall usage of the backbone is task-dependent and highly related to how much the tasks share the learned knowledge. Hence, overall sparsity on the backbone increases naturally yet sparsity per task will be the same for each task.

**Sparse Initialization.** We consider the following strategies for sparse initialization of the network.

*Uniform*: Sparsity of each layer, denoted as $s$, is equivalent to the overall network sparsity, represented by $S$.

*Erdős-Rényi Kernel (ERK)* [31]: Inspired by Erdős-Rényi initialization [28], ERK is formulated as $\frac{\epsilon(n^{l-1}+n^l+w^l+h^l)}{n^{l-1}n^l w^l h^l}$ where $\epsilon$ indicates the sparsity level, $n^l$ denotes the number of neurons at layer $l$, $w^l$ and $h^l$ are the width and the height of the $l^{th}$ convolutional kernel. It allocates higher sparsities to the layers with more parameters and lower sparsities to the layers with fewer parameters while keeping the total sparsity at the level of $S$. The sparsity of the fully connected layers follows the same rules of the original Erdős-Rényi.

**Topology Update.** The topology update should balance *exploration vs. exploitation* trade-off to obtain better sparse topologies. The update schedule is determined by the following parameters: (1) $\Delta T$, the number of iterations between sparse connectivity updates (2) $T_{end}$, the iteration at which the sparse connectivity updates should be stopped (3) $\alpha$, the initial proportion of connections that are updated, and (4) $f_{decay}$, a function that is called every $\Delta T$ iteration which gradually decreases the proportion of updated connections over time. Similar to the previous studies [25, 31, 32, 34], we use Cosine Annealing $f_{decay}(t, \alpha, T_{end}) = \frac{\alpha}{2}(1 + \cos(\frac{t\pi}{T_{end}}))$ for topology update.

**Drop and Growth.** We consider the following drop and growth strategies to update the topology.

*Magnitude-based drop* [37]: It involves identifying and removing the connections with the lowest magnitudes or 'saliency' based on a predefined sparsity level or threshold.

*Random growth* [28]: It randomly adds new connections to the network and does not take into account any specific information about the network's behavior or performance.

*Unfired growth*: It randomly adds new connections that have not been used or 'fired' before.

*Gradient growth* [31]: It involves adding new connections based on the gradients values of the connections during the training process. Connections with higher gradients or those that contribute more to the network's error reduction are considered for growth.

*Momentum growth* [32]: It considers both the current gradient and the accumulated gradient from previous updates which smoothes out the updates over time. Connections with higher momentum are considered for growth since they are more promising to reduce the network's error.

Gradient and momentum growth techniques are able to accelerate the search for an optimal subnetwork but they necessitate a dense backward pass that incorporates information from both existing and non-existing connections.

# 4. Experimental Setup

In this section, we briefly describe our experimental setup including the datasets used, the metrics employed for evaluation, and the implementation details that ensure the reproducibility and validity of our results.

**Datasets.** In this paper, we want to create scenarios with different numbers of tasks to see the impact of the DST strategies under various conditions. To this end, we experiment with CIFAR100 [38] and miniImageNet [39] which consist of objects from 100 different categories. We split CIFAR100 into 5, 10 and 20 distinct tasks with 20, 10 and 5 classes within each learning task and named them **5-task CIFAR100**,

**10-task CIFAR100**, and **20-task CIFAR100** respectively. Finally, for a more challenging scenario, we use **10-task miniImageNet** in which the model learns 10 tasks with 10 classes within each incremental step.

**Metrics.** We resort to the well-known CL metrics for evaluation. **Accuracy (ACC)** [40] measures the final accuracy averaged over all tasks where $A_{T,i}$ is the accuracy of task $i$ after learning task $T$. **Backward transfer (BWT)** [40] measures the level of forgetting and the average accuracy change of each task after learning new tasks where $A_{i,i}$ is the accuracy of task $i$ just after learning task $i$. **Forward transfer (FWT)**, inspired from [40], measures the level of transfer where $A_{j,i}$ is the accuracy of learned subnetwork on task $i$ performing on task $j$, evaluates how well the model performs on the following tasks which are different from the tasks it was trained on.

$$ACC = \frac{1}{T} \sum\nolimits_{i=1}^{T} A_{T,i} \tag{2}$$

$$BWT = \frac{1}{T-1} \sum\nolimits_{i=1}^{T-1} (A_{T,i} - A_{i,i}) \tag{3}$$

$$FWT = \frac{1}{T-i} \sum\nolimits_{i=1}^{T} \sum\nolimits_{j=i+1}^{T} A_{j,i} \tag{4}$$

**Implementation Details.** We implement all the methods in PyTorch [41]. We use ResNet-18 [42], VGG-like [43], and MobileNetV2 [44] as the backbone for all datasets. For VGG-like and MobileNetV2 results, please refer to the Appendix. We choose not to freeze batch normalizations during the training because they cannot be assigned as task-specific components. We have built on the ITOP library [34] which is developed for standard batch learning. We use Cosine Annealing for a drop rate starting from 0.5. We update the sparse topology after every 50, 100, 200, and 400 iterations. We set number of epochs to 25 for 20-task CIFAR100 whereas 100 for 5-task CIFAR100, 10-task CIFAR100 and 10-task miniImageNet. We use adam optimizer and set an initial learning rate of 0.001 which is decayed by a factor of 0.1 after 1/2 and 3/4 of the total number of epochs. We set the batch size to 128. We run experiments on three different seeds and report their average.

## 5. Results

In this section, we present the empirical analysis to investigate the effect of different DST components in a CL paradigm to understand the underlying mechanism and shed light on the optimal DST configuration for CL. We study on four distinct scenarios with different incremental steps and datasets which are 5-task CIFAR100, 10-task CIFAR100, 20-task CIFAR100, and 10-task miniImageNet to observe the effect of initialization, growth, and its adaptive selection between tasks. We also study the topology update frequency in the Appendix.

### 5.1. Different initialization strategies require different sparsity levels.

In combination with various growth techniques and sparsity levels, initialization strategy significantly influences the incremental accuracy, especially when the number of tasks increases and tasks become more complex such as 10-Task miniImageNet, 10-Task CIFAR100 and, 20-Task CIFAR100. At the high sparsity level unless it is extreme (see Appendix), we found that uniform initialization performs more stable under different growth strategies compared to ERK. Although the existing parameter isolation studies in continual learning [25, 26] have primarily employed uniform initialization, which aligns with our first finding, we also intriguingly find that ERK initialization exhibits good overall performance under low to moderate sparsity level (Figure 2, 3, 4, 5). In particular, uniform initialization saturates after 7 tasks, while the ERK initialization successfully completes all 10 tasks (Figure 3a, 3d and Figure 4a, 4d). The same phenomenon is also observed with 20-Task CIFAR100 where uniform initialization saturates after 7 tasks again, while the ERK initialization reaches this saturation after 13 tasks which indicates 85% efficient backbone usage (Figure 2a, 2d).

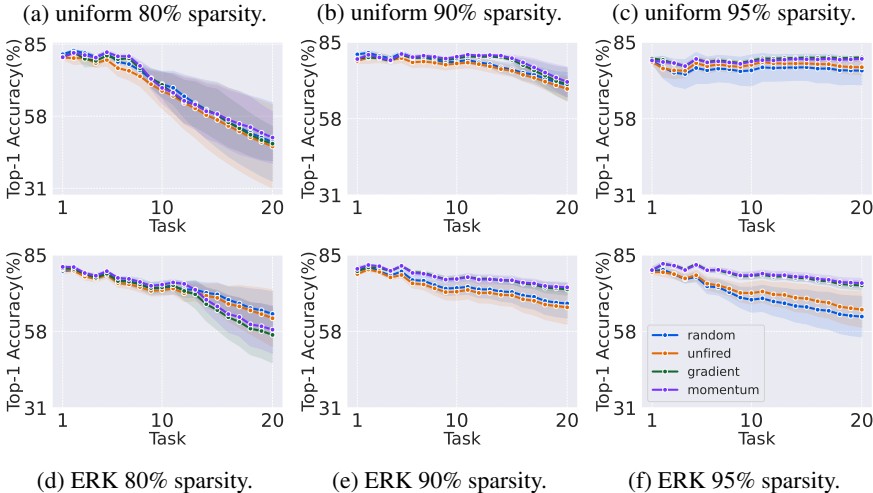

|  | (a) uniform 80% sparsity. | (b) uniform 90% sparsity. | (c) uniform 95% sparsity. |
|  | (d) ERK 80% sparsity. | (e) ERK 90% sparsity. | (f) ERK 95% sparsity. |

Figure 2: Top-1 ACC (%) of 20-Task CIFAR100 with different DST strategies. At a low to moderate sparsity, ERK utilizes the backbone more efficiently by using the connections within the narrow layers sparingly which allows to effectively learn longer increments of tasks and improves incremental performance: Uniform saturates after 7 tasks when sparsity level per task is 80%. ERK reaches this saturation after 13 tasks. At the high sparsity, however, uniform shows a more reliable and robust performance.

Table 1: Top-1 ACC (%), BWT (%), and FWT (%) scores of 20-Task CIFAR100.

| sparsity | 0.8 | | | 0.9 | | | 0.95 | | |
|---|---|---|---|---|---|---|---|---|---|
| uniform | ACC | BWT | FWT | ACC | BWT | FWT | ACC | BWT | FWT |
| random | 52.70 ±17.86 | 0.01±0.28 | 14.74 ±3.78 | 69.91 ±5.05 | **1.56 ±3.26** | **18.61 ±0.38** | 74.96 ±5.81 | **0.17 ±0.24** | 18.52 ±0.98 |
| unfired | 46.67 ±18.03 | **0.13 ±0.38** | 13.53 ±3.93 | 69.44±5.63 | -0.34 ±0.25 | 18.55 ±0.27 | 76.15 ±3.09 | -0.10 ±0.36 | 18.73 ±0.55 |
| gradient | 47.71 ±7.30 | -0.23 ±0.42 | 12.80 ±1.80 | 70.17 ±6.65 | -0.73 ±0.24 | 18.02 ±1.17 | **79.60 ±0.59** | 0.07 ±0.37 | **18.96 ±0.25** |
| momentum | 50.01 ±11.12 | 0.18 ±0.36 | 13.58 ±2.92 | 70.97 ±6.08 | 0.04 ±0.33 | 18.23 ±0.67 | 79.12 ±0.97 | -0.21 ±0.10 | 18.87 ±0.16 |
| sparsity | 0.8 | | | 0.9 | | | 0.95 | | |
| ERK | ACC | BWT | FWT | ACC | BWT | FWT | ACC | BWT | FWT |
| random | **63.42 ±10.23** | -0.12 ±0.36 | **14.87 ±1.93** | 67.87 ±5.99 | 0.01 ±0.30 | 17.08 ±1.10 | 63.24 ±8.25 | -0.26 ±0.29 | 18.27 ±1.10 |
| unfired | 61.97 ±14.02 | -0.08 ±0.19 | 14.64 ±2.73 | 66.51 ±6.92 | -0.19 ±0.08 | 16.74 ±1.13 | 65.74 ±12.41 | -0.10 ±0.08 | 18.55 ±0.35 |
| gradient | 56.77 ±11.37 | -0.13 ±0.12 | 13.23 ±2.69 | 72.85 ±2.39 | 0.01 ±0.65 | 17.50 ±0.79 | 74.23 ±1.92 | 0.01 ±0.35 | 18.72 ±0.15 |
| momentum | 60.35 ±11.12 | -0.32 ±0.47 | 14.20 ±2.25 | **73.57 ±1.96** | -0.05 ±0.62 | 17.81 ±0.32 | 75.07 ±1.94 | 0.08 ±0.13 | 18.81 ±0.26 |

The reason behind this is the fact that uniform initialization creates a bottleneck within the narrow layers. We know that uniform initialization assigns equal connections to each layer without considering the layer size and ERK allocates more connections to the layers with more parameters and less connections to the layers with fewer parameters [28, 31]. Following a certain number of tasks, backbone's capacity for sustaining the learning process is compromised since information flow through the backbone gets drastically decreased for the subsequent tasks. Our findings provide empirical support for the relationship between sparsity and information flow in CL setup: Specifically, in scenarios where the sparsity levels are high, we observe that information flow remains sustainable even with the uniform initialization. This suggests that the network retains sufficient connectivity to effectively accommodate new tasks. However, as the sparsity decreases, indicating a higher density of connections assigned per task, we observe a noticeable challenge in learning future tasks when the backbone is initialized uniformly. Therefore, under the same conditions, ERK initialization is better at protecting information flow by using the connections within the narrow layers sparingly (see also Appendix). Note that the reason to have these small BWT (%) scores is freezing all the connections except the batch normalization as we stated and discussed in the *Implementation Details*.

## 5.2. Growth strategy affects performance, tied to initialization and sparsity levels.

Our experimental results indicate that when the sparsity level is low, ERK initialization with random and unfired growth outperforms other alternatives (Table 1, 2, 3, 4). While the sparsity level is moderate, ERK and uniform initialization achieve similar performance where uniform initialization combined with gradient growth (uniform+gradient) tends to show a slightly better performance (Table 2, 3). In the case of high sparsity, the common approach of uniform+gradient [26] achieves noticeable performance while the ERK+gradient

combination also demonstrates competitive results (Table 2, 3, 4). However, at an high sparsity level, the ERK+random and ERK+unfired exhibit lower accuracy performance (Table 1, 2, 3, 4). We also observe that uniform initialization is more resilient to the choice of growth strategies compared to ERK initialization (Figure 2, 3, 4, 5).

Overall, for the growth strategies in DST, gradient- and momentum-based growth provide better performance compared to random and unfired growth in the case of high sparsity yet random and unfired growth perform as well as gradient- and momentum-based growth in low-moderate sparsity. The rationale behind this is gradient and momentum-based methods are able to incorporate more information and can selectively add new connections between neurons that allow for faster exploration compared to random or unfired growth which needs more time to explore and find better connections.

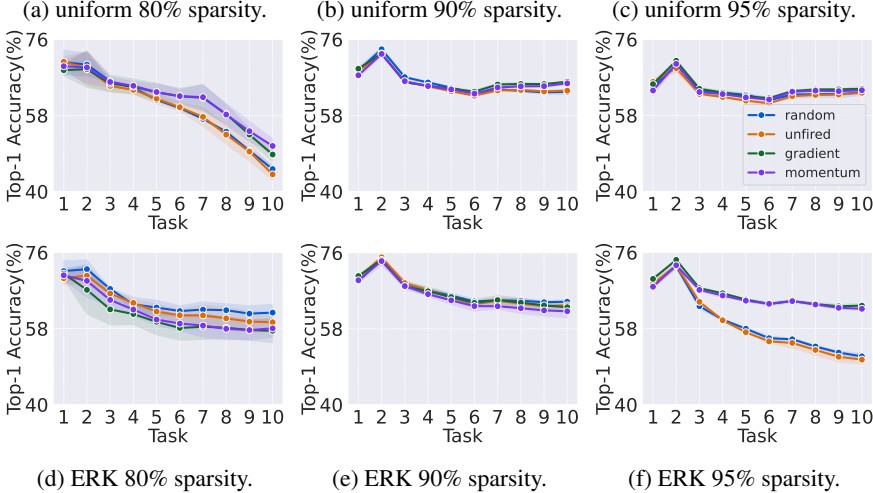

Figure 3: Top-1 ACC (%) of 10-Task miniImageNet with different DST criteria. ERK works better at low sparsity: Uniform saturates after 7 tasks while ERK completes all tasks. In moderate sparsity, all DST approaches exhibit highly comparable performance. In the higher sparsity level, uniform emerges as a solid approach to work with different growths but it is worth mentioning that ERK is able to perform nearly identical if right growth is selected.

Table 2: Top-1 ACC (%), BWT (%), and FWT (%) scores of 10-Task miniImageNet.

| sparsity | 0.8 | | | 0.9 | | | 0.95 | | |
|---|---|---|---|---|---|---|---|---|---|
| uniform | ACC | BWT | FWT | ACC | BWT | FWT | ACC | BWT | FWT |
| random | 45.32 ±1.39 | **0.42±0.67** | 9.54 ±0.51 | 64.19 ±0.85 | -0.08 ±0.08 | 9.38 ±0.34 | 63.43 ±1.01 | -0.18 ±0.29 | 9.36 ±0.08 |
| unfired | 44.06 ±1.85 | 0.26 ±0.63 | 9.06 ±0.71 | 63.87±0.22 | -0.29 ±0.81 | 9.47 ±0.16 | 63.35 ±0.04 | -0.27 ±0.67 | 9.41 ±0.08 |
| gradient | 48.75 ±2.36 | -0.05 ±0.06 | 8.57 ±0.44 | **65.97 ±0.18** | 0.11 ±0.47 | **9.61 ±0.01** | **64.29 ±0.79** | -0.21 ±0.94 | 9.65 ±0.25 |
| momentum | 50.78 ±2.41 | 0.21 ±0.19 | 8.81 ±0.24 | 65.59 ±0.24 | -0.48 ±0.11 | 9.39 ±0.03 | 63.96 ±0.71 | 0.15 ±0.24 | 9.29 ±0.11 |
| sparsity | 0.8 | | | 0.9 | | | 0.95 | | |
| ERK | ACC | BWT | FWT | ACC | BWT | FWT | ACC | BWT | FWT |
| random | **61.78 ±2.72** | -0.25 ±0.65 | **9.75 ±0.01** | 64.41 ±0.46 | 0.26 ±0.44 | 9.54 ±0.13 | 51.43 ±1.07 | **0.39 ±0.11** | **9.72 ±0.27** |
| unfired | 59.49 ±3.21 | -0.46 ±1.58 | 9.27 ±0.43 | 63.55 ±0.32 | 0.06 ±0.79 | 9.54 ±0.12 | 50.66 ±1.41 | -0.22 ±0.41 | 9.22 ±0.41 |
| gradient | 57.47 ±3.01 | 0.15 ±0.73 | 8.31 ±0.48 | 63.12 ±1.38 | 0.05 ±0.23 | 9.28 ±0.17 | 63.47 ±0.68 | -0.02 ±0.22 | 9.38 ±0.01 |
| momentum | 58.06 ±3.98 | -0.46 ±0.44 | 8.39 ±0.34 | 62.15 ±2.25 | **0.36 ±0.02** | 8.78 ±0.38 | 62.68 ±0.46 | 0.37 ±0.19 | 9.16 ±0.25 |

Table 3: Top-1 ACC (%), BWT (%), and FWT (%) scores of 10-Task CIFAR100.

| sparsity | 0.80 | | | 0.90 | | | 0.95 | | |
|---|---|---|---|---|---|---|---|---|---|
| uniform | ACC | BWT | FWT | ACC | BWT | FWT | ACC | BWT | FWT |
| random | 58.04 ±1.96 | -0.03 ±0.26 | 9.03 ±0.27 | 75.09 ±0.91 | -0.13 ±0.56 | 9.35 ±0.16 | 74.39 ±0.17 | -0.02 ±0.22 | 8.97 ±0.23 |
| unfired | 56.03 ±0.57 | -0.16 ±0.18 | 8.41 ±0.29 | 75.28 ±0.94 | 0.025 ±0.27 | 9.27 ±0.15 | 74.38 ±0.22 | -0.10 ±0.09 | 9.22 ±0.21 |
| gradient | 63.29 ±1.72 | 0.002 ±0.03 | 9.06 ±0.19 | **76.63 ±0.41** | -0.11 ±0.17 | 9.49 ±0.12 | **75.57 ±0.21** | -0.19 ±0.16 | **9.56 ±0.09** |
| momentum | 62.92 ±1.12 | 0.03 ±0.19 | 8.94 ±0.09 | 76.49 ±0.26 | **0.11 ±0.30** | 9.26 ±0.23 | 75.40 ±0.44 | -0.07 ±0.32 | 9.18 ±0.12 |
| sparsity | 0.80 | | | 0.90 | | | 0.95 | | |
| ERK | ACC | BWT | FWT | ACC | BWT | FWT | ACC | BWT | FWT |
| random | 74.61 ±0.44 | -0.05 ±0.22 | 9.52 ±0.24 | 75.71 ±0.32 | -0.06 ±0.11 | **9.68 ±0.25** | 65.81 ±0.37 | **-0.01 ±0.36** | 9.49 ±0.12 |
| unfired | **74.86 ±0.54** | 0.025 ±0.41 | **9.52 ±0.22** | 75.33 ±0.33 | -0.06 ±0.07 | 9.64 ±0.14 | 65.56 ±0.49 | -0.16 ±0.24 | 9.46 ±0.29 |
| gradient | 70.63 ±1.91 | **0.14 ±0.31** | 7.91 ±0.28 | 74.71 ±2.33 | 0.01 ±0.42 | 8.68 ±0.14 | 74.81 ±0.37 | -0.04 ±0.25 | 8.97 ±0.13 |
| momentum | 69.51 ±3.26 | -0.29 ±0.25 | 7.57 ±0.55 | 73.73 ±2.43 | 0.02 ±0.31 | 8.21 ±0.74 | 74.14 ±0.22 | -0.03 ±0.25 | 8.44 ±0.13 |

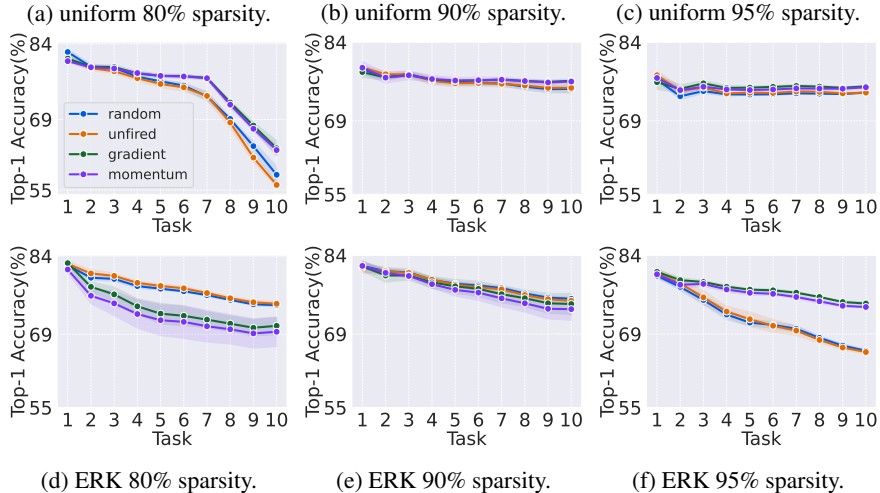

Figure 4: Top-1 ACC (%) of 10-Task CIFAR100 with different DST criteria. Findings align with the 10-Task miniImageNet. At the high sparsity, uniform appears as a more reliable strategy with respect to different growths yet ERK is able to perform nearly identical if right growth is selected.

Table 4: Top-1 ACC (%), BWT (%), and FWT (%) scores of 5-Task CIFAR100.

| sparsity | 0.80 | | | 0.90 | | | 0.95 | | |
|---|---|---|---|---|---|---|---|---|---|
| uniform | ACC | BWT | FWT | ACC | BWT | FWT | ACC | BWT | FWT |
| random | 68.37 ±0.58 | -0.16 ±0.09 | 4.54 ±0.09 | 67.92 ±0.93 | 0.04 ±0.23 | 4.36 ±0.19 | 65.68 ±0.62 | 0.01 ±0.12 | 4.33 ±0.11 |
| unfired | 68.79 ±0.71 | **0.13 ±0.33** | 4.53 ±0.18 | 67.57 ±0.93 | 0.04 ±0.28 | 4.42 ±0.11 | 65.69 ±0.72 | **0.15 ±0.33** | 4.35 ±0.29 |
| gradient | 70.21 ±0.55 | -0.36 ±0.51 | 4.74 ±0.21 | 69.43 ±0.52 | 0.17 ±0.17 | 4.86 ±0.23 | **68.25 ±0.31** | 0.05 ±0.33 | **5.12 ±0.18** |
| momentum | 69.75 ±1.23 | -0.025 ±0.32 | 4.46 ±0.22 | 67.93 ±0.46 | -0.14 ±0.16 | 4.21 ±0.13 | 65.12 ±0.97 | -0.24 ±0.14 | 4.15 ±0.41 |
| sparsity | 0.80 | | | 0.90 | | | 0.95 | | |
| ERK | ACC | BWT | FWT | ACC | BWT | FWT | ACC | BWT | FWT |
| random | 73.72 ±0.09 | -0.16 ±0.36 | 5.19 ±0.21 | 72.7 ±0.33 | -0.13 ±0.29 | 5.13 ±0.05 | 61.7 ±1.32 | -0.22 ±0.26 | 4.36 ±0.39 |
| unfired | **73.73 ±0.27** | 0.03 ±0.11 | **5.34 ±0.21** | **73.01 ±0.27** | **0.18 ±0.08** | **5.17 ±0.17** | 61.86 ±2.41 | -0.11 ±0.31 | 4.11 ±0.35 |
| gradient | 64.76 ±3.34 | -0.12 ±0.33 | 3.58 ±0.26 | 68.87 ±1.47 | 0.02 ±0.25 | 3.85 ±0.03 | 67.92 ±0.38 | -0.08 ±0.24 | 3.66 ±0.14 |
| momentum | 64.16 ±2.37 | -0.14 ±0.19 | 3.51 ±0.14 | 68.43 ±1.17 | -0.01 ±0.31 | 3.45 ±0.09 | 66.53 ±0.73 | 0.06 ±0.22 | 3.23 ±0.14 |

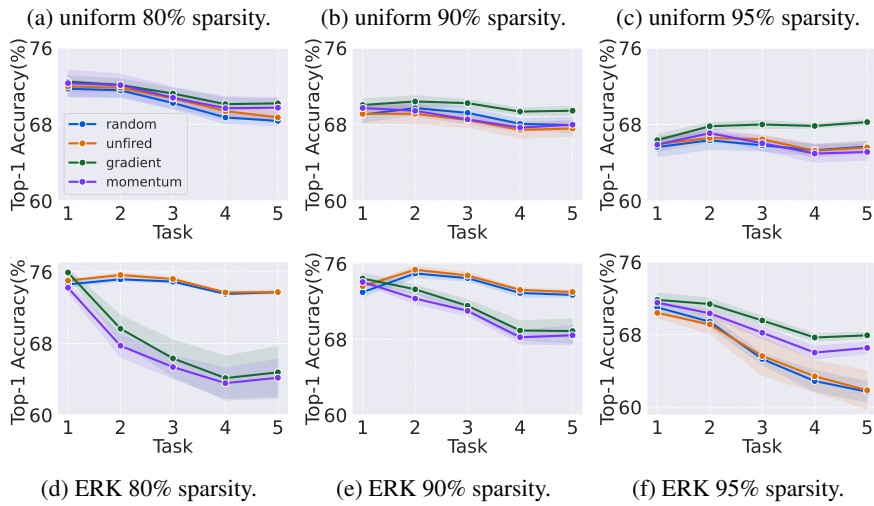

Figure 5: Top-1 ACC (%) of 5-Task CIFAR100 with various DST strategies. Interestingly, findings align with 20-Task CIFAR100: ERK improves incremental accuracy at a low to moderate sparsity. On the other hand, at the high sparsity, uniform appears as a more reliable strategy yet ERK is able to perform nearly identical if right growth is selected.

### 5.3. Adaptivity improves performance since there is no panacea.

Throughout our experiments, we have observed that there is no universally successful approach in terms of different task numbers and sparsity levels. When the number of tasks is high, the choice of both initialization and growth strategies become crucial, depending on the sparsity level. It is unlikely to exist a panacea DST setup for all datasets or tasks, and it is not feasible to try different alternatives for each scenario due to computational cost and time. Therefore, we hypothesized that an adaptive approach for selecting DST criteria per task should improve the performance.

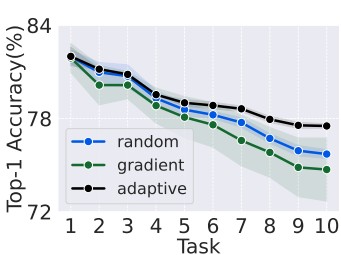

Figure 6: Adaptive growth strategy on 10-task CIFAR100 scenario. Adaptive approach outperforms the fixed strategy.

In our final experiment, we construct a naive adaptive growth strategy with ERK initialization to validate our hypothesis, wherein random growth is implemented for the first five tasks, and gradient growth is selected for the subsequent five tasks while sparsity per task is 90% on the 10-task CIFAR100 dataset. The adaptive approach yields better performance when compared to two baselines which are pre-selected and fixed growth strategies of random and gradient with ERK initialization (Figure 6). This is because of that during the initial stage of learning, there exists considerable potential for exploration of the model parameters. Therefore, a random growth strategy tends to be more efficient and effective when the exploration space is sufficiently large. However, as the model capacity becomes more saturated, the gradient growth strategy becomes increasingly effective and superior to random growth which results in better overall performance.

## 6. Discussion

**Conclusion.** This paper has presented a comprehensive analysis, aiming to evaluate different components of Dynamic Sparse Training in Continual Learning. We focus on the Parameter Isolation strategy to find the best subnetwork for each task under various conditions that highly affect the final topologies. Through large-scale experiments, we found that choice of sparsity level, initialization, and growth strategy significantly affects the incremental performance. In our findings, when sparsity is low, Erdős-Rényi Kernel (ERK) initialization exploits the backbone efficiently, enabling the learning continually. However, unless the sparsity is extreme, uniform initialization demonstrates more consistent and stable performance at higher sparsity levels. The performance of the growth strategy relies on the chosen initialization method and the level of sparsity. As a final observation, even naive adaptive selection for DST criteria per task improves the incremental performance compared to using fixed and pre-defined strategies for all tasks. We hope empirical results that are found in this study inspire the community to make informed decisions regarding the applicability of these strategies in real-world scenarios and pave the way for future studies in this domain.

**Limitations and Future Work.** To extend the scope of this study, structured dynamic sparse training should also be analyzed to grasp the underlying mechanism of dynamic sparse training comprehensively in continual learning. Finally, the observed performance improvements in incremental learning are significant with a simple adaptive approach. Therefore, further investigation on more sophisticated adaptive DST strategies promises a highly promising research direction.

## Acknowledgements

This work is supported by; TAILOR, a project funded by EU Horizon 2020 research and innovation programme under GA No. 952215, Dutch national e-infrastructure with the support of SURF Cooperative using grant no. EINF-4568, and Turkish MoNE scholarship.

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

# Appendix

In this appendix, we provide additional insights into the effect of different topology update frequencies, sparsity levels, and models in the context of continual learning with dynamic sparse training.

**Update Frequencies.** When the sparsity level is lower, a topology update frequency of 400 iterations demonstrates optimal performance. This suggests that with more connections, updating the network topology less frequently is more advantageous. Conversely, when the sparsity is higher, around 90-95%, a more frequent topology update interval of 100-200 iterations appears to be more beneficial (Figure A). This observation can be attributed to the increased need for exploration in densely connected networks. More frequent updates allow the model to adapt and explore new connections more effectively in order to accommodate the changing data distribution. These findings underscore the importance of adapting topology update strategies based on the level of sparsity during continual learning, offering potential avenues for further optimization and efficiency in dynamic sparse training methods.

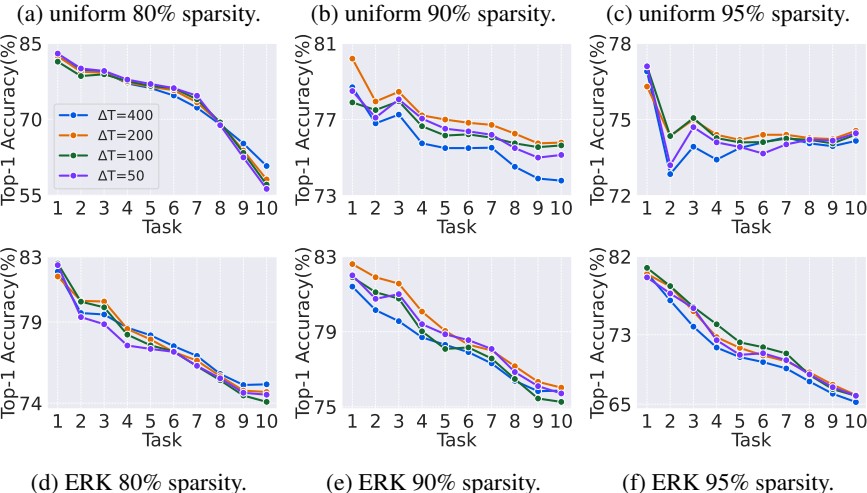

Figure A: Top-1 ACC (%) of 10-Task CIFAR100 with different initializations and topology update frequencies when growth strategy is selected as random; using ResNet-18. Note that y-scale is different for each plot to demonstrate differences clearly.

**Sparsity Levels.** Our findings pointed towards the superiority of ERK initialization at lower to moderate sparsity levels, while uniform initialization appeared more effective at higher sparsity settings. Upon conducting additional experiments with sparsity levels of 70% and 85%, we reiterate that ERK initialization consistently outperforms uniform initialization. However, it is noteworthy that when we extend our analysis to an extreme sparsity level of 99% where the network operates under significantly constrained conditions ERK initialization once again prove to be the more suitable choice. That is because the limitation of uniform initialization becomes evident in continual learning when sparsity reaches extreme levels. This constraint hinders the model's ability to effectively discover multiple subnetworks, which, in turn, diminishes its capacity to consistently represent complex functions. The root of this issue lies in uniform initialization's approach to distributing connections uniformly across layers. This leads to an excessively reduced information flow, preventing the model to continuously representing complex functions. Consequently, uniform initialization is only able to produce a single proficient subnetwork, while hampering the information flow and adaptability of the main backbone for evolving requirements of sequential tasks. On the other hand, by allocating higher sparsities to the layers with more parameters and lower sparsities to the layers with fewer parameters, ERK allows to craft of subnetworks that effectively enable the continuity of information flow across successive tasks. This underscores the crucial need for adaptability in choosing the right initialization method, particularly when dealing with varying sparsity requirements, to ensure the success of continual learning systems.

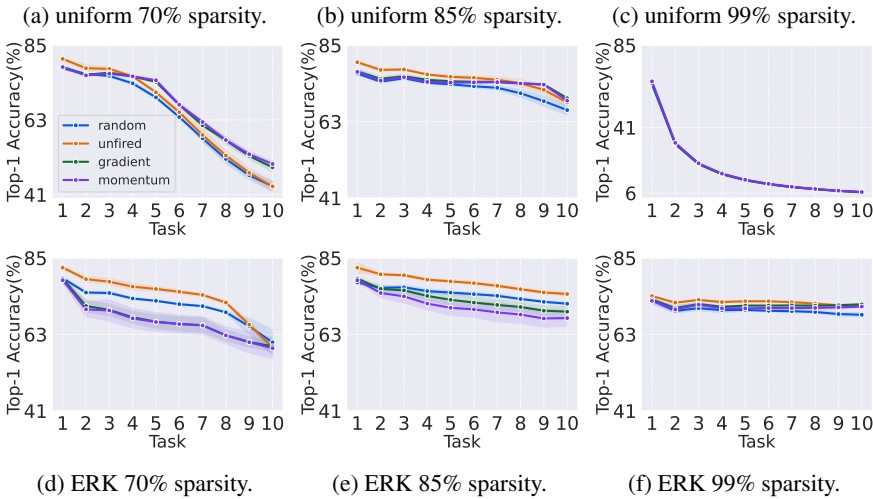

Figure B: Top-1 ACC (%) of 10-Task CIFAR100 with different initialization, growth, and sparsity levels; using ResNet-18. In extreme sparsity case, a significant drop occurs on uniform initialization.

**Different Models.** Additional experimental results, using MobileNetV2 and VGG-like architectures, are presented in Figure C and D. For MobileNetV2, a compact network known for its efficiency, we explore sparsity levels of 70%, 80%, and 90%. Our observations highlight the consistent effectiveness of both uniform and ERK initialization when the sparsity levels are 70% and 80%. In the 90% sparsity, which is an extreme for MobileNetV2, we find parallel findings with our previous experiments: Uniform initialization encounters a familiar challenge; it demonstrates limited learning capability for the first task while negatively impacting the information flow and adaptability to the demands of sequential tasks.

For VGG-like architecture, we explore sparsity levels of 80%, 90%, and 95%. All initializations are performed well since the VGG-like network is large enough to protect the information flow while learning more than one task sequentially. However, the incremental performance depends on a growth strategy at all sparsity levels.

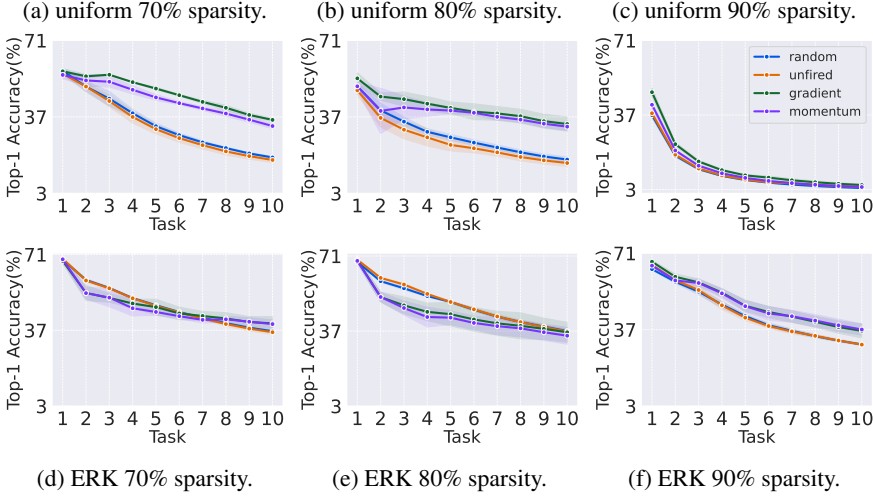

Figure C: Top-1 ACC (%) of 10-Task CIFAR100 with different initialization, growth and sparsity levels; using MobileNetV2.

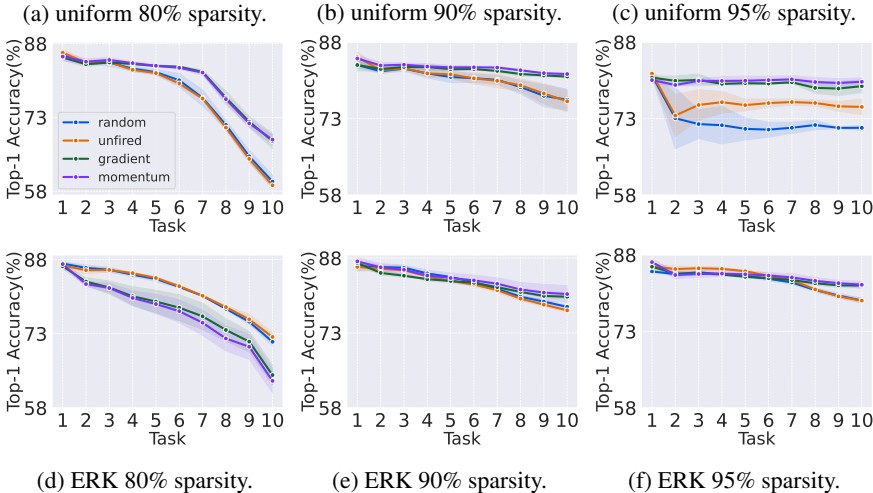

(a) uniform 80% sparsity.    (b) uniform 90% sparsity.    (c) uniform 95% sparsity.

(d) ERK 80% sparsity.    (e) ERK 90% sparsity.    (f) ERK 95% sparsity.

Figure D: Top-1 ACC (%) of 10-Task CIFAR100 with different initialization, growth and sparsity levels; using VGG-like.

