# OpenReview forum: "Continual Learning with Dynamic Sparse Training: Exploring Algorithms for Effective Model Updates"
_CPAL.cc/2024/Conference — CPAL 2024 (Proceedings Track) Oral_

### Official Review · Reviewer_v5pG · 2023-10-07
**Please see the review**

**Rating:** 6
**Confidence:** 4

**Review:**

This paper proposes a comprehensive analysis, aiming to evaluate different components of Dynamic Sparse Training(DST) in Continuous Learning (CL).  Extensive experiments are conducted to support the findings.

Pros:

1. The work of evaluating the different settings of DST in CL scenarios can be a guide for later research.

2. The experiments on the different initialization of DST and the prune-and-grow approach are comprehensive.

Cons & Question

1. Do the findings proposed in this paper help increase the performance of previous work like SparCL, NISPA, and WSN?  It would be great to have some comparisons with other methods.

2. The findings are based on ResNet structure. Can similar results be found in VGG and MobileNet?

3. It is interesting that adaptively choosing DST methods can improve performance, as shown in 5.3. Does this finding exist in all the sparsity levels (Low sparsity and high sparsity)?

4. Most of the findings apply to DST (e.g. ERK distribution can reach higher accuracy, and gradient- and momentum-based growth provide better performance), and I would like to know what is uniquely specific to CL.

5. As ITOP systematically evaluate the impact of topology update frequency, I am curious about the effect of update frequency on CL.

---

### Official Review · Reviewer_YxBF · 2023-10-07
**please see the review.**

**Rating:** 6
**Confidence:** 4

**Review:**

The paper investigates the use of Dynamic Sparse Training (DST) strategies in Continual Learning (CL) scenarios. The authors explore different initialization and growth strategies and their impact on performance. They conduct experiments on CIFAR100 and miniImageNet datasets with varying sparsity levels and task numbers. The choice of initialization and growth strategies depends on the sparsity level and number of tasks. The adaptive approach proposed by the authors shows promising results in enhancing performance.

Advantage:

1.	The paper explores the performance of different DST strategies in the context of continual learning (CL) tasks.

2.	It proposes an adaptive approach for selecting DST criteria per task, which improves performance compared to fixed strategies.

Disadvantage:

1.	It's unclear how the DST strategies discussed in the paper perform when applied to other neural networks like VGG and MobileNet. Additional experimentation or information on their applicability to different architectures could be valuable.

2.	The paper focuses on sparsity levels of 80%, 90%, and 95% to draw conclusion that at a low to moderate sparsity level, ERK initialization is more efficient and at a high sparsity level, uniform initialization is more robust. It would be beneficial to know how these strategies perform at other sparsity levels, like higher than 95%, lower than 80%, or more fine-grained sparsity levels between 80% and 95%. Also, it would be better to know how these strategies perform on other works, will other works demonstrate the same conclusion?

3.	I wonder if the proposed adaptive method can be applied to other works, will the performance of adaptive method be better than random or gradient in other works?

4.	It would be better to show how the frequency of topology updates affects performance.

---

### Official Review · Reviewer_dCsQ · 2023-10-07

**Rating:** 5
**Confidence:** 5

**Review:**

Overview

This paper investigating the effect of different DST components under the CL paradigm by performing a comprehensive study in which we investigate various DST components to find the best topology per task on well-known CIFAR100 and miniImageNet benchmarks in a task-incremental CL setup.


Strengths

1. It is very interesting results that ERK and uniform sparsity experience different performance at different sparsity levels under CL setting. The results have practical significance to the further research on CL.

2. Adaptivity within DST is also evaluated under CL setting, and the design is very important in practical use of DST, not just in CL.



Weaknesses

1. The observation is sufficient, however, there is no detailed discussion on why the DST has such difference in CL in terms of its settings on sparsity.

2. The main body of the paper should have more discussion on the rationale of DST’s different settings.

---

### Meta-Review · Area_Chair_CW2g · 2023-11-14

**Recommendation:** Accept (Poster)
**Confidence:** 4

**Metareview:**

This submission provides a valuable empirical analysis of Dynamic Sparse Training (DST) in Continual Learning (CL), contributing notably to the field. The study's methodical examination of DST components using CIFAR100 and miniImageNet benchmarks is commendable. The findings regarding the performance of Erdos-Renyi Kernel (ERK) and uniform initialization across different sparsity levels are insightful. While further exploration of various neural architectures and a deeper theoretical discussion could enhance the paper, its solid methodology and relevance stand out. Overall, the paper is a worthy addition to CPAL 2024.

---

### Decision · Program_Chairs · 2023-11-20

**Decision:**

Accept (Oral)

**Comment:**

This paper investigates the effect of different dynamic sparse training components under the continual learning paradigm. The reviewers and AC agree that overall, the analysis in the paper is a valuable contribution to this research area. The authors perform a comprehensive study in which they investigate various dynamic sparse training components to find the best topology per task in a task-incremental continual learning setup. The reviewers generally found the experiments in the paper convincing in support of the claimed findings. However, some reviewers raised concerns about the applicability of the proposed approach to other neural networks and the performance of the proposed approach at different sparsity levels. The authors should consider addressing these concerns in the camera ready version of the paper.

The action PC chair for this paper is Gintare Karolina Dziugaite, who made the decision after carefully reading the paper as well as the comments by all reviewers and AC. The decision is agreed by all PC chairs.